# Sequencing of SARS-CoV-2 RNA Fragments in Wastewater Detects the Spread of New Variants during Major Events

**DOI:** 10.3390/microorganisms11112660

**Published:** 2023-10-30

**Authors:** Damir Zhakparov, Yves Quirin, Yi Xiao, Nicole Battaglia, Michael Holzer, Martin Bühler, Walter Kistler, Doortje Engel, Jon Paulin Zumthor, Alexa Caduff, Katja Baerenfaller

**Affiliations:** 1Swiss Institute of Allergy and Asthma Research (SIAF), University of Zurich, 7265 Davos, Switzerland; damir.zhakparov@uzh.ch (D.Z.);; 2Swiss Institute of Bioinformatics (SIB), 1005 Lausanne, Switzerland; 3Cantonal Office for Nature and Environment, 7000 Chur, Switzerland; 4Cantonal Office for Food Security and Animal Health, 7000 Chur, Switzerland; 5Cantonal Office for Military and Civil Protection, 7000 Chur, Switzerlandalexa.caduff@amz.gr.ch (A.C.); 6Hospital Davos, 7270 Davos, Switzerland; 7Cantonal Office for Health, 7000 Chur, Switzerland

**Keywords:** amplicon sequencing, COVID-19 pandemic, epidemiological surveillance, relative prevalence, SARS-CoV-2, variant mapping, wastewater

## Abstract

The sequencing of SARS-CoV-2 RNA in wastewater is an unbiased method to detect the spread of emerging variants and to track regional infection dynamics, which is especially useful in case of limited testing and clinical sequencing. To test how major international events influence the spread of new variants we have sequenced SARS-CoV-2 RNA in the wastewater samples of Davos, Landquart, Lostallo, and St. Moritz in the Swiss canton of Grisons in the time around the international sports competitions in Davos and St. Moritz in December 2021, and additionally in May 2022 and January 2023 in Davos and St. Moritz during the World Economic Forum (WEF) in Davos. The prevalence of the variants identified from the wastewater sequencing data showed that the Omicron variant BA.1 had spread in Davos and St. Moritz during the international sporting events hosted there in December 2021. This spread was associated with an increase in case numbers, while it was not observed in Landquart and Lostallo. Another instance of new variant spread occurred during the WEF in January 2023, when the Omicron variant BA.2.75 arrived in Davos but not in St. Moritz. We can therefore conclude that major international events promote the spread of new variants in the respective host region, which has important implications for the protective measures that should be taken.

## 1. Introduction

At the beginning of 2020 the wild-type strain of the severe acute respiratory syndrome coronavirus-2 (SARS-CoV-2) began to spread worldwide, causing the outbreak of coronavirus disease 2019 (COVID-19). SARS-CoV-2, originally termed new coronavirus (2019-nCoV) due to its sequence similarity with SARS-CoV, was identified and characterized in early 2020. SARS-CoV-2 is one of the seven known human coronaviruses, and together with SARS-CoV and MERS-CoV, which can cause severe pneumonia, it belongs to the genus *Betacoronavirus*. Due to its high sequence similarity with SARS-CoV-2-related CoVs found in various bat species, it likely originates from bats [1,2,3]. SARS-CoV and SARS-CoV-2 not only exhibit considerable sequence similarity but also both use angiotensin converting enzyme II (ACE2) as a cellular entry vector [2,4]. ACE2 is not only expressed in epithelial cells of the bronchi and the oral mucosa, but also in the intestine and the colon [5,6]. RT-PCR measurements in the stool samples of patients with COVID-19 revealed high concentrations of SARS-CoV-2 RNA [7,8,9]. Viral shedding in feces and the detection of SARS-CoV-2 RNA in wastewater brought up the interesting possibility of using wastewater monitoring to understand the epidemiology of COVID-19 [10,11]. Wastewater surveillance has therefore subsequently gained a lot of attention during the pandemic because it provides a way to obtain early and objective information about virus circulation independent of testing, which is inherently subject to bias. The main method for analyzing wastewater samples is to quantify viral gene counts by quantitative PCR after the enrichment of viral RNA from the samples. As different factors such as wastewater dilution, changes in the number of inhabitants, and the amount of human fecal material will skew the quantification of the SARS-CoV-2 gene count, the data are often normalized for flow rate per capita or by the quantity of surrogate viruses [12,13,14]. Another interesting method is to sequence SARS-CoV-2 RNA directly from the wastewater samples either by nested RT-PCR or targeted amplification and amplicon sequencing, followed by the determination of viral lineages and their relative prevalence [15,16,17,18,19,20,21]. The wastewater sequencing of viral RNA and variant mapping therefore provide valuable information about the presence and prevalence of viral variants, which complements clinical sequencing efforts.

For the first months of the COVID-19 pandemic, SARS-CoV-2 showed limited apparent evolution due to non-pharmaceutical interventions such as lockdowns that limited the global virus population and the spread of the virus. However, since late 2020 a set of mutations, which impact virus characteristics such as antigenicity or infectivity, emerged in the context of variants of concern (VOCs). The first three divergent lineages with a high number of mutations emerging in different regions of the world were Alpha, Beta and Gamma. These distinct lineages exhibit some convergent mutations, implying that they confer a fitness advantage in evading population immunity, which subjects the virus to selective pressure. The following period was characterized by a pronounced evolutionary diversification marked by a gradual increase in divergence within the major lineages combined with a stepwise increase as new major lineages emerged [22,23,24]. On 26 November 2021, the World Health Organization (WHO) named a newly emerging VOC, Omicron, which was highly mutated compared to the other VOCs. Soon after the first Omicron variant BA.1 had spread, it was replaced by BA.2 and BA.5. BA.2 then diversified into different sub-lineages including BA.2.75, and BA.5 into BQ.1.1, among others. These newer Omicron variants evolved under increased immune selection in immunized and previously infected populations compared to the first VOCs and carry additional mutations that are likely responsible for increased immune evasion and infectivity [25,26,27,28,29].

The genomic epidemiology of SARS-CoV-2 heavily relies on the sequencing data of the viral genome produced in laboratories all over the world and deposited in the GISAID platform with restricted access to data [30] and/or fully open databases such as GenBank [31]. These sequencing data are analyzed by the Nextstrain bioinformatics project to produce phylogenies that show the evolutionary relationships of SARS-CoV-2 viruses [32] and by the CoVariants project to provide an overview of the variants and the mutations that are of interest [33]. This information now forms the basis for assigning the amplicon sequencing data of viral RNA fragments isolated from wastewater samples to the individual SARS-CoV-2 variants [15]. This allows the local spread of emerging variants of SARS-CoV-2 to be detected at the population level using the genomic sequencing of wastewater samples in a time series. For instance, this enabled the monitoring of a local outbreak of the Alpha variant in wastewater samples from a Swiss ski resort during the holiday season in December 2020 [15].

The Swiss canton of Grisons has a complex geography with around 150 valleys, and towns, villages, and regions with locally very different population dynamics. In the main tourist destinations, the peak holiday season between December and March brings great population dynamics with a large temporary influx of people from different places. Major events including international sports competitions or the World Economic Forum (WEF) annual meeting reinforce this effect and temporarily multiply the number of accommodated persons in the respective host regions. The alpine cities Davos and St. Moritz hosted two major sports events on the weekend of 11 to 12 December 2021. In St. Moritz, it was the Women’s FIS Alpine Ski World Cup, and in Davos, the International cross-country days with the FIS Nordic World Cup. While the first half of December is not yet peak tourist season, travel activities had already increased in the weeks before the sports events due to the requirement for altitude acclimatization because Davos is 1560 m over sea level, and St. Moritz 1822, respectively. Another event that attracts many international visitors to Davos is the annual WEF, which took place from 22 to 26 May in 2022 and from 16 to 20 January in 2023. In order to detect new spreading strains and observe how tourism, travel and population dynamics around the time of the abovementioned major events can influence the spread of new variants, we decided to monitor the prevalence of different viral variants by sequencing viral RNA fragments isolated from wastewater samples. These samples were taken from the wastewater treatment plants Davos and S-chanf, into which St. Moritz and other villages in the Oberengadin valley drain. In November and December 2021, additional wastewater samples were taken for comparison from the less crowded village of Landquart, with lower population fluctuations, and from Lostallo, because of its proximity to the Swiss canton of Ticino and the Italian border.

## 2. Materials and Methods

### 2.1. Sample Collection

Wastewater was collected from the four wastewater treatment plants S-chanf, Davos, Landquart, and Lostallo in the Swiss canton of Grisons in November and December 2021, and from S-chanf and Davos in May 2022 and January 2023. The 24-h composite samples were kept at 4 °C during transport and processed centrally in the cantonal laboratory.

### 2.2. Extraction of Total Nucleic Acid from Wastewater Samples

Total nucleic acid was isolated by applying the direct capture method [34] using the Maxwell RSC Enviro TNA Kit (Promega, Dübendorf, Switzerland) following the manufacturer’s instructions. In brief, 40 mL of a 24-h composite influent wastewater sample was treated with 0.5 mL of protease solution and incubated for 30 min at RT. After centrifugation at 3000× *g* for 10 min at RT, the supernatant was transferred to a glass bottle and gently mixed with 12 mL of binding buffer 1 and 1 mL of binding buffer 2. After adding 48 mL of isopropanol, the samples were passed through a PureYield™ Midi Binding Column using a VacMan^®^ Vacuum Manifold (Promega, Switzerland) The columns were washed by adding 5 mL of column wash buffer 1 followed by 20 mL of column wash buffer 2. Bound nucleic acids were eluted by adding 2 × 300 µL of pre-heated (60 °C) nuclease free water. To further purify the sample, 600 µL of the eluate was mixed with 150 μL of Binding Buffer 1 and 50 μL of Binding Buffer 2. The solution was transferred to the Maxwell^®^ RSC Cartridge and nucleic acids were eluted automatically in 80 μL of nuclease-free water. The final concentrate was used for sequencing.

### 2.3. Sequencing

Sequencing libraries were prepared from the wastewater RNA extracts of the 2021 samples using the COVID-19 ARTIC V3 protocol in which the reverse transcribed cDNA was amplified with the ARTIC V3 panel as described previously [15]. For the 2022 and 2023 samples, the SARS-CoV-2 ARTIC V4.1 protocol was used [35]. The amplicons were end-repaired and polyadenylated and ligated to adaptors. The amplicon fragments containing adapters on both ends were enriched and barcoded using the amplicon barcoding protocol. The libraries were sequenced producing paired-end-reads of 250 bp in length using the Illumina NovaSeq 6000 and MiSeq platforms.

### 2.4. Variant Mapping

All the sequencing files in .fastq format were downloaded. For the analysis of the short-read sequencing data of the SARS-CoV-2 RNA fragments from the 2021 samples, the rubicon branch of the V-pipe Github repository was downloaded [36], and the raw data were processed with V-pipe for quality control, alignment, and generation of base counts. The variant mutation list was built based on the definition of variants that is listed under the voc folder of the public repository (https://github.com/cbg-ethz/cojac/tree/4344e81f8050d2b09deaef4aaca400141ecf6dc9/voc, update of 27 January 2022). The following variants were included in the variant list: B.1.1.7 (Alpha), B.1.351 (Beta), B.1.617.1 (Kappa), B.1.617.2 (Delta), BA.1 and BA.2 (Omicron), and P.1 (Gamma). In the COJAC section of V-pipe, a tally of mutation occurrences was done based on the signatures provided [15]. In brief, the base count results were gathered and variant-characteristic mutations in the samples were identified according to the mutation list. The relative frequency of each signature mutation was determined, and all frequencies were combined, providing an estimate of the relative prevalence of the variant in the population. For each location, 100 mutations were randomly drawn with replacement from all the mutations that occurred on a specific day. The prevalence for each variant was estimated using Ridge regression, and lowess smoothing was applied to generate the curves over time. The resampling and estimation process was repeated 100 times and a 95% confidence interval was calculated based on the resampling distribution. For the 2022 and 2023 samples, the ninjaturtles branch of V-pipe was used following the steps described with some minor adjustments (https://github.com/cbg-ethz/V-pipe/tree/ninjaturtles, accessed on 12 July 2023) [15] and with a variant definition that was based on the mutations provided in the V-pipe Github repository for the SARS-CoV-2 variants (https://github.com/cbg-ethz/cowwid/tree/master/voc, accessed on 12 July 2023). The deconvolution of these samples to quantify the prevalence of each variant was performed with the Lollipop workflow (https://github.com/cbg-ethz/LolliPop, accessed on 12 July 2023) [37], which performs the following: (1) it creates a design matrix to map mutations to the respective virus variant definitions; (2) based on the loss function, it aligns observed mutation frequencies to compute relative abundances of each variant; (3) to account for temporal changes, it applies kernel smoothing to the data; and (4) finally, the uncertainty is estimated by two approaches in which analytical confidence intervals are calculated and then bootstrapping is applied to calculate the empirical confidence intervals. The preset setting deconv_bootstrap_cowwid.yaml for Lollipop was taken from the Github repository as specified in the usage guidelines. The prevailing variants at the time of wastewater sampling were obtained from the CovSpectrum website (https://academic.oup.com/bioinformatics/article/38/6/1735/6483076, accessed on 12 July 2023). The prevalence plots were generated using the ggplot2 R package. All files with the bash scripts and settings files are available at https://github.com/dzhakparov/wastewater_publication/tree/master, accessed on 12 July 2023.

## 3. Results

To monitor how major events affect the prevalence of different variants of SARS-CoV-2 and the spread of emerging variants, we collected 24-h composite wastewater samples from wastewater treatment plants in the Swiss canton of Grisons in the time around the FIS Alpine and Nordic World Cup races in November and December 2021, and around the WEF in May 2022 and January 2023. After RNA extraction, the ARTIC amplicon-based targeted genome sequencing approach was used to determine the sequence of the SARS-CoV-2 fragments. Using these data, the prevalence of the individual variants was determined from the set of variants that occurred at the respective time points using the V-pipe pipeline [36].

### 3.1. Spread of Omicron BA.1 during International Sports Competitions in December 2021

Wastewater samples were collected from the wastewater treatment plants in Landquart on 5, 12, 19, 26 November and 10 and 23 December, in Lostallo on 5, 19, 26 November and 3, 10, and 23 December, and in S-chanf and Davos on 5, 12, 19, 26 November and 3, 5, 7, 9, 10, 12, 14, 16, 19, 21, and 23 December. The sequencing of the SARS-CoV-2 RNA fragments and the determination of the prevalence of the prevailing variants revealed that the variant B.1.617.2 (Delta) was dominant at all four sites in early November. Other circulating variants on 5 November 2021 in Davos and S-chanf were B.1.617.1 (Kappa) (4.4% in Davos, 3.9% in S-chanf) and B.1.1.7 (Alpha) (2.8% in Davos, 2.6% in S-chanf). The emerging first Omicron variant BA.1 reached a relative prevalence of larger than 5% in Davos and S-chanf on 2 December 2021, but not in Landquart and Lostallo. In Davos and S-chanf the relative prevalence of BA.1 was steadily rising and reached 66.1% in Davos and 49.6% in S-chanf on 23 December 2021, becoming the dominant variant. At this date, BA.1 was basically not present in Landquart and Lostallo, with relative prevalences of less than 1% (Figure 1, Appendix A).

The arrival of Omicron BA.1 was also monitored in the clinical sample sequencing data according to CoVariants [33]. The strain identified in all 322 clinical samples sequenced during the period of 8 and 22 November 2021 in Region 6 of Switzerland, encompassing the four testing sites included here, remained the Delta variant. In the period between 22 November and 6 December 2021, three out of two hundred and fifty two sequenced strains were Omicron BA.1 (21K), which increased to 76 out of 242 sequenced samples between 6 and 20 December 2021. In the next period between 20 December 2021 and 3 January 2022, already 410 of 477 sequenced strains were BA.1, reaching a frequency of 86% (Table 1).

### 3.2. The Increased Relative Prevalence of Omicron BA.1 Caused Higher Infection Rates

As part of its protection concept, from the beginning of 2021, the canton of the Grisons adopted a strategy for symptom-free mass testing, where people could be PCR-tested for an infection with SARS-CoV-2 free of charge on a weekly basis [38]. On 1 June 2021, almost 2000 companies with a total of over 25,000 employees and 165 schools with around 23,000 pupils were already taking part in weekly company tests. This means that the canton of Grisons tested about a third of its mobile population every week, which allowed the exact development of the pandemic to be monitored.

While the number of positive PCR tests per day resulted in highly variable data, the number of active cases, where a person who tested positive was counted as actively infected for 10 days, resulted in data that better matched the infection dynamics. The number of active cases from 5 November to 23 December 2021 in the catchment areas of the four wastewater treatment plants Davos, Landquart, Lostallo, and S-chanf revealed different infection dynamics. In Landquart, the number of active cases peaked on 25 November with 297 cases, but then gradually decreased. In Lostallo, the number of infected persons was very low in November and only slightly increased in December to 24 active cases. In contrast, the number of active cases in Davos and in the catchment area of the S-chanf wastewater treatment plant where St. Moritz is located, started to increase towards the end of November and on 23 December reached a maximum of 155 active cases in Davos and of 141 active cases in S-chanf, respectively (Figure 2). In these two locations the increasing number of active cases therefore matches well with the increasing prevalence of Omicron BA.1. Considering the higher infectivity of Omicron BA.1 compared to the wild-type strain and the other circulating variants of concern at the time [27,29], this suggests that the increased infection rates here were caused by the increase in the prevalence of Omicron BA.1.

### 3.3. No Effect of the WEF in May 2022 on the Relative Prevalence of the Dominant Omicron Variants

Due to the spread of the Omicron variant in late 2021, WEF 2022, which was originally scheduled to take place from 17 to 21 January, has been postponed to 22 to 26 May 2022. For the wastewater sequencing around the time of the WEF, samples were taken on 15, 19, 22, 24, 26, and 29 May 2022 at the wastewater treatment plants S-chanf and Davos. In Davos, the two predominant Omicron variants on 15 May 2022 were already BA.5 with a relative prevalence of 62.3% and BA.2 with 37.3%. This pattern essentially remained until 29 May, when BA.5 was at 67.7% and BA.2 at 31.2%. In S-chanf, on 15 May the relative prevalence of the dominant BA.2 was at 98.3%, and BA.5 arrived only later. On 19 May the relative prevalence of BA.5 was at 35.6% and increased to 58.3% on 29 May when it became the dominant variant (Figure 3, Appendix A). At the time around WEF 2022 we therefore observed basically no change in the relative prevalence of Omicron variants BA.2 and BA.5 in Davos but monitored the potential arrival of BA.5 in S-chanf. Due to the discontinuation of the mass testing in early 2022, we cannot assess how this was related to the number of active cases in the two locations.

### 3.4. Arrival and Spread of Omicron BA.2.75 during WEF 2023 in Davos

WEF reverted to January in 2023 and took place from 16 to 20 January. For the wastewater sequencing, samples were collected from the wastewater treatment plants S-chanf and Davos on 12, 15, 17, 19, 22, and 24 January. Before the WEF on 12 January, the Omicron variants XBB.1.5 (43.2%), XBB (21.7%), XBB.1.9 (20.1%), and BA.5 (11.3%) were circulating in Davos. In S-chanf, it was essentially the Omicron variants XBB.1.5 (55.8%) and XBB (41%). The prevalence distribution changed considerably in Davos from 17 January on, when Omicron variant BA.2.75 arrived at a relative prevalence of 3.8%, which then increased to 44% on 22 January when it became the dominant variant. In contrast, XBB.1.5 and XBB remained the dominant variants in S-chanf with BA.2.75 only arriving at a relative prevalence of maximum 2.2% on 17 January (Figure 4, Appendix A).

Before the WEF, in the period from 2 to 9 January 2023, the Omicron variant BA.2.75 was predominantly observed in Asia, Australia, and New Zealand, with frequencies exceeding 50% in Vietnam, Thailand, Georgia, and South Korea (Table 2). When examining clinical sample sequencing data from Region 6 in Switzerland, as in CoVariants [32], no BA.2.75 (22D) was detected among the 45 sequenced samples in the time period between 19 December 2022 and 2 January 2023. In the subsequent time period, between 2 and 16 January 2023, already 2 out of 36 sequenced samples were found to be BA.2.75 (Table 3).

The official number of participants at WEF 2023 was 2653 people from business, politics, science, and culture from 127 different countries, most of whom came from Northern America (761), followed by Western Europe (551), Northern Europe (338) and Western Asia (213) (Table 4 and Appendix A). In addition to those officially participating in the WEF, there were tens of thousands of other visitors, some of whom belonged to international companies that use the WEF for representation purposes. This temporarily increased the population of Davos by about five times. Not surprisingly, this massive influx lead to a local spread of the SARS-CoV-2 variants circulating worldwide. In this scenario, the most contagious and immunoevasive variants are expected to have an advantage.

## 4. Discussion

Wastewater sampling is an effective and unbiased method to obtain population-level information on various aspects such as the consumption of pharmaceuticals or drugs or, as during the COVID-19 pandemic, epidemiological dynamics. The method of the wastewater sequencing of SARS-CoV-2 RNA and the determination of variant prevalence has been widely adopted worldwide. Even at the beginning of the pandemic when wild-type SARS-CoV-2 was spreading, wastewater sequencing was used to assess viral diversity in samples from 11 states in the USA and from 4 different wastewater treatment plants in the San Francisco Bay Area, and it was found that the wastewater data identified more circulating lineages than represented in the clinical data [17,39]. Comparing the results acquired using RT-PCR screening and genome sequencing in wastewater and using patient sample sequencing in Marseille, France revealed that sequencing allows for the detection of a distribution of variants in wastewater that is similar to patient sample sequencing, while some variants are missed by RT-PCR [40]. In Spain, sequencing of SARS-CoV-2 RNA in wastewater from 14 wastewater treatment plants during three epidemiological waves allowed for the detection of the Alpha variant and showed the capacity of the method to detect mutations before they are detected in clinical samples [41]. In addition, in Switzerland, the Alpha variant was detected in wastewater up to 13 days before it was first reported in clinical samples and wastewater sequencing was found to provide population-level estimates for the prevalence of emerging variants earlier and based on fewer samples than based on clinical samples [15]. The wastewater monitoring initiative at the University of California San Diego (UCSD) campus in the USA also revealed that emerging VOCs could be identified up to 14 days earlier in wastewater samples in comparison to clinical genomic surveillance. Furthermore, it was observed that instances of virus transmission could be detected in wastewater samples that had not been identified through clinical testing [21]. Furthermore, pan-European wastewater-based SARS-CoV-2 surveillance comparing the mutation profiles of Alpha, Beta, Gamma, and Delta in 54 European municipalities revealed that data on SARS-CoV-2 VOCs in wastewater samples mirror variant profiles obtained from clinical data [42]. These studies have highlighted the ability of wastewater sequencing to identify circulating SARS-CoV-2 variants within defined geographical regions delineated by the catchment area of the respective wastewater treatment plants, either independently or in addition to clinical genomic surveillance data.

The alternative approach to analyzing wastewater samples by quantifying viral gene counts by quantitative PCR was also successfully applied in the Swiss canton of Grisons from October 2020 on. However, after the emergence of Omicron, the testing of wastewater samples with PCR no longer provided reliable information on the SARS-CoV-2 gene count, because some of the primer pairs used to detect the wild-type strain and the earlier VOCs did not recognize Omicron. Therefore, the PCR protocols had to be redesigned so that they could also recognize Omicron (Appendix A). As the time period of the first testing interval of this study coincided with the arrival of Omicron, the focus here was therefore not on quantifying viral gene counts, but on determining the relative prevalence of different SARS-CoV-2 variants. The prevalence of different variants can also be determined with the sequencing of clinical samples. In addition, these viral genome sequence data are integral for mapping variants in amplicon sequencing data [32]. In the clinical sample sequencing data according to CoVariants [33], BA.1 was first detected in the canton of Grisons and Ticino in the time period between 22 November and 6 December 2021 with a frequency of 1.2%. In view of these data, the early peak of BA.1, which reached a relative prevalence of 10.4% on 26 November 2021 in Landquart in the data presented here (Figure 1, Appendix A), might be an artifact of the mapping procedure. Further supporting this view are the additional peaks in the relative prevalences of the Kappa and Alpha variants, as well as in the undetermined fraction, which was only observed in the Landquart samples on those dates. These findings suggest the potential lower data quality of these samples, which could contribute to the overdispersion effect. The quantification section of the V-Pipe pipeline [15] uses statistical modeling to detect and quantify the presence of SARS-CoV-2 variants. Since smoothing regression is utilized to assess the prevalence of every variant, it is prone to estimation mistakes when there are not enough neighboring data points or when the sequence data are not of sufficient quality. It can also be misled when variants are included in the analyses, which are no longer or not yet circulating at some of the time points. In the analyses that were carried out during the three different test periods, it was therefore extremely important that the lists of the prevailing variants were used for mapping.

The increasing relative prevalence of Omicron BA.1 in November and December 2021 matched with the increasing numbers of active cases in Davos and in the catchment area of the S-chanf wastewater treatment plant as determined by the PCR testing of patients. This can most probably be attributed to the higher infectivity of BA.1 compared to the, at the time, dominant Delta variant [27,29]. The higher infectivity of BA.1 also had considerable consequences on the cantonal testing strategy. The success of the symptom-free mass testing strategy was based on the concept that the early detection and isolation of infected persons combined with contact tracing results in the interruption of infection chains [38,43]. Since this was no longer the case with the higher infectivity of Omicron, mass testing was discontinued when Omicron became dominant. As the SARS-CoV-2 gene count assessment in the wastewater samples became unreliable with the arrival of a highly mutated new variant until PCR protocols were re-established, this source of information was also no longer available. This demonstrates the value of RNA amplicon sequencing followed by variant mapping as an unbiased methodology in pandemic surveillance, as this method generates sequencing data that remain valid and can be reanalyzed using updated variant definitions when novel VOCs arrive.

Wastewater sequencing data could also prove to be particularly useful in scenarios characterized by limited clinical sample sequencing capacity. In comparison to clinical testing during the period from late December to January in 2021/2022, the year 2022/2023 witnessed a stark decline, with only approximately one-tenth of the samples being sequenced within the same timeframe. This reduced testing volume is likely to introduce a notable bias, as it is probable that only the more severe cases will undergo sequencing. Consequently, the arrival of new variants that exhibit higher infectivity but a similar risk of hospitalization may go unnoticed, potentially disrupting timely protective measures and strategies. We therefore intend to maintain wastewater monitoring during future large international events, despite its discontinuation here as a routine practice.

In terms of the early implementation of protective measures, the canton of Grisons played an important pioneering role at the beginning of the pandemic. For instance, the cantonal government took proactive measures by canceling the Engadin Ski Marathon and other events on 27 February 2020, even before COVID-19 was declared a pandemic on March 11 by the WHO [44]. In retrospect, and taking into account the data presented here, which demonstrate that large events can contribute to the dissemination of novel virus variants, these early and resolute decisions were well-founded in their aim to contain the outbreak of the pandemic within a population lacking immunity to SARS-CoV-2.

## Figures and Tables

**Figure 1 microorganisms-11-02660-f001:**
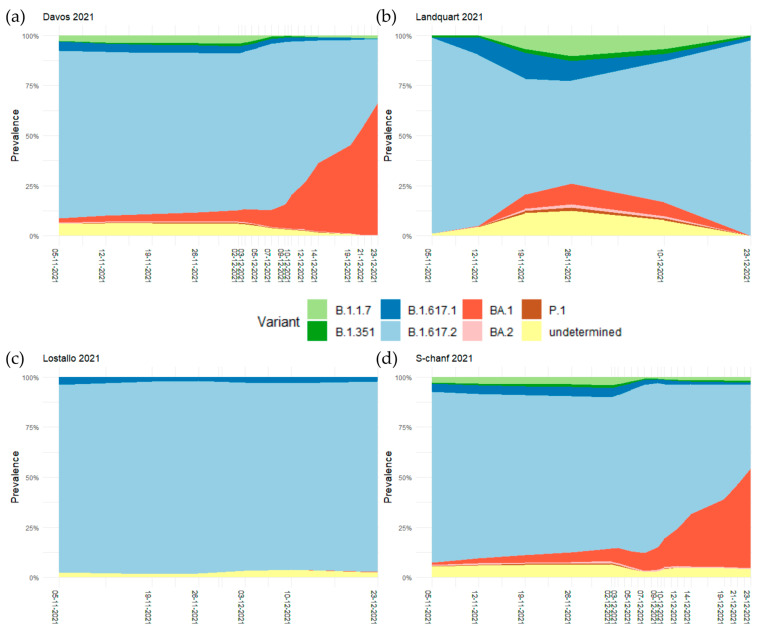
Smoothed regression curves of the relative prevalence of the SARS-CoV-2 variants B.1.1.7 (Alpha), B.1.351 (Beta), B.1.617.1 (Kappa), B.1.617.2 (Delta), BA.1 and BA.2 (Omicron), and P.1 (Gamma) in November and December 2021 in (**a**) Davos, (**b**) Landquart, (**c**) Lostallo, and (**d**) S-chanf.

**Figure 2 microorganisms-11-02660-f002:**
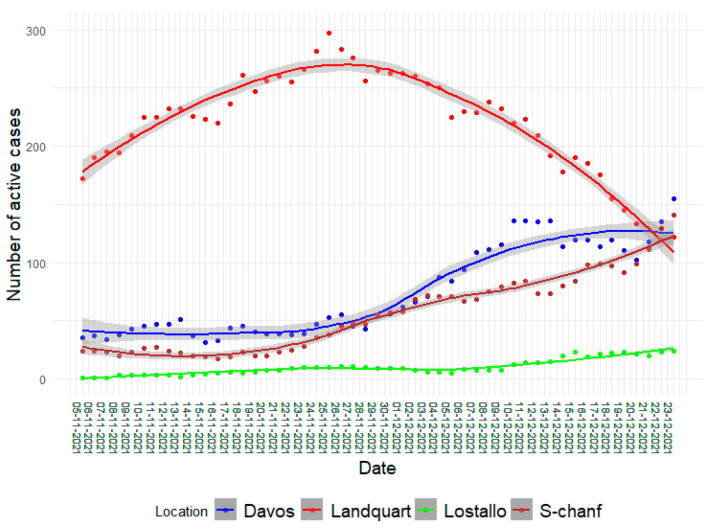
The number of active cases in the catchment area of the Davos, Landquart, Lostallo, and S-chanf wastewater treatment plants from 5 November to 23 December 2021 with a smoothed curve using the Loess method. The number of active cases is calculated by counting the positively tested infected persons as active for 10 days from the day of the positive PCR test.

**Figure 3 microorganisms-11-02660-f003:**
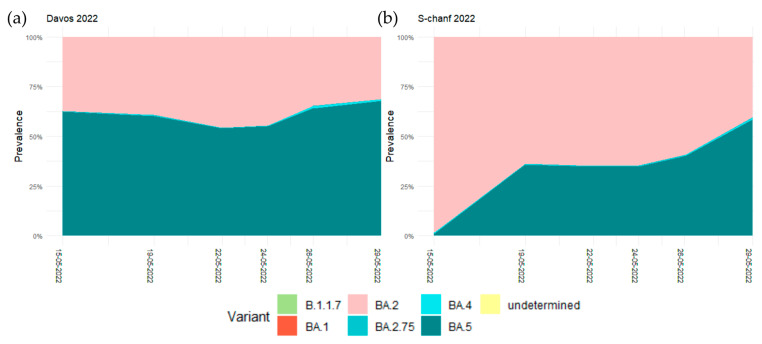
Smoothed regression curves of the relative prevalence of the SARS-CoV-2 variants B.1.1.7 (Alpha) and the Omicron variants BA.1, BA.2, BA.2.75, BA.4, and BA.5 in May 2022 in (**a**) Davos and (**b**) S-chanf.

**Figure 4 microorganisms-11-02660-f004:**
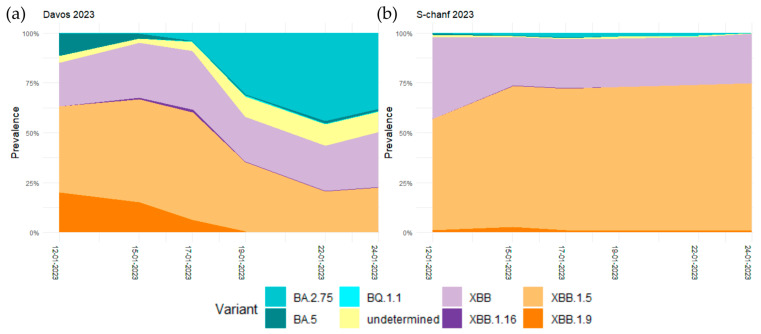
Smoothed regression curves of the relative prevalence of the SARS-CoV-2 Omicron variants BA.2.75, BA.5, BQ.1.1, XBB, XBB.1.16, XBB.1.15, XBB.1.19 in January 2023 in (**a**) Davos and (**b**) S-chanf.

**Table 1 microorganisms-11-02660-t001:** Results of the clinical sample sequencing data according to CoVariants [33] in the four time periods between 8 November 2021 and 3 January 2022 in Region 6 of Switzerland.

8 November 2021–22 November 2021		
Variant	Num seq	Frequency
21J (Delta)	307	0.95
21I (Delta)	14	0.04
21A (Delta) (B.1.617.2)	1	0.00
22 November 2021–6 December 2021		
Variant	Num seq	Frequency
21J (Delta)	241	0.96
21I (Delta)	8	0.03
21K (Omicron) (BA.1)	3	0.01
6 December 2021–20 December 2021		
Variant	Num seq	Frequency
21J (Delta)	163	0.67
21K (Omicron) (BA.1)	76	0.31
21I (Delta)	3	0.01
20 December 2021–3 January 2022		
Variant	Num seq	Frequency
21K (Omicron) (BA.1)	410	0.86
21J (Delta)	66	0.14
21I (Delta)	1	0.00

**Table 2 microorganisms-11-02660-t002:** The frequency of BA.2.75 between 2 and 9 January 2023 in the 20 countries with the highest frequency according to their geographical region as in CoVariants [32].

Asia	Eastern Asia	South Korea	0.53
		Mauritius	0.45
	South-eastern Asia	Viet Nam	0.92
		Thailand	0.84
		Cambodia	0.44
		Singapore	0.19
		Malaysia	0.14
	Western Asia	Georgia	0.65
		Qatar	0.28
Europe	Eastern Europe	Romania	0.20
		Bulgaria	0.19
		Czech Republic	0.17
		Slovakia	0.15
	Northern Europe	Lithuania	0.22
		Iceland	0.18
		Denmark	0.17
	Southern Europe	Greece	0.15
		Croatia	0.12
Oceania	Australia/ New Zealand	Australia	0.33
		New Zealand	0.17

**Table 3 microorganisms-11-02660-t003:** Results of the clinical sample sequencing data according to CoVariants [33] in the two time periods between 19 December 2022 and 16 January 2023 in Region 6 of Switzerland.

19 December 2022–2 January 2023		
Variant	Num seq	Frequency
22E (Omicron) (BQ.1)	32	0.71
22B (Omicron) (BA.5)	11	0.24
others	1	0.02
23C (Omicron) (CH.1.1)	1	0.02
2 January 2023–16 January 2023		
Variant	Num seq	Frequency
22E (Omicron) (BQ.1)	26	0.72
22B (Omicron) (BA.5)	4	0.11
22F (Omicron) (XBB)	2	0.06
22D (Omicron) (BA.2.75)	2	0.06
23C (Omicron) (CH.1.1)	1	0.03
21L (Omicron) (BA.2)	1	0.03

**Table 4 microorganisms-11-02660-t004:** The official number of WEF 2023 participants per region of origin.

Africa	139	North Africa	20
		Eastern Africa	38
		Middle Africa	6
		Southern Africa	52
		Western Africa	23
Americas	881	Caribbean	4
		Central America	28
		South America	88
		Northern America	761
Asia	622	Central Asia	2
		Eastern Asia	184
		South-eastern Asia	106
		Southern Asia	117
		Western Asia	213
Europe	992	Eastern Europe	32
		Northern Europe	338
		Southern Europe	71
		Western Europe	551
Oceania	19	Australia/New Zealand	18
		Melanesia	1

## Data Availability

The data presented in this study are openly available in the SRA database and can be found here: https://www.ncbi.nlm.nih.gov/sra/PRJNA1011526.

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
