# Peer review of "Sequencing of SARS-CoV-2 RNA Fragments in Wastewater Detects the Spread of New Variants during Major Events"

_microorganisms, 2023, doi:10.3390/microorganisms11112660_

Round 1
Reviewer 1 Report
Comments and Suggestions for Authors
Congratulations on a well-written manuscript. You have a simple and straightforward design for your study. Wastewater-based surveillance is a great method for unbiased community-wide public health surveillance. However, it serves the public better when it complements clinical data. As more data is produced through wastewater-based epidemiology it will become a standalone surveillance system. Your study lacks originality. It is not clear what novel, wet or dry, laboratory methodology was used to produce the results obtained. The manuscript reads like a surveillance report. I have made few comments below to help increase confidence in the results you have presented.
Introduction. The background given for the VOC and the the description of the sampling area were well written and appropriate to the context of the study. There was not enough background given regarding wastewater-based sequencing in general and of SARS-CoV-2 in particular. The first paragraph of the discussion gives a nice background that helps understand the aim of the study and should be part of the introduction instead. Please add a reference for the sentence going from line 58 to 60.
Research design. The design of the project is appropriate for the aim of the study. The sampling dates and locations were well chosen. The sampling dates for the 2021 sport events occurred at least 10 days before the event and 10 days after. The fact that you have searched for defining mutations of VOCs instead of consensus sequences increase confidence in your data. You have made used of the ARTIC v3 protocol for the 2021 samples, and, ARTIC v4.1 for 2022 and 2023 samples. Amplicons drop off were observed when using the v3 primer set. Did you compare the two protocols on your wastewater samples before applying it? What novelty does your study bring? With regard to the use of wastewater-based sequencing; sample processing? data analysis?
Methodology. Well described. Please revise the following:
The back and forth use of S-chanf and St-Moritz between the text and the graphs is confusing. I understand that St-Moritz is located in the catchment area of S-chanf. Choose one and keep it uniform throughout the text and figures to facilitate comparison. Given that sampling is done at S-chanf, but the events happened at St-Moritz description can be given in the introduction or sample collection then S-chanf kept throughout the text and on figures.
line 91: you mentioned collection from three wastewater treatment plants, but cited 4, i.e. S-chanf, Davos, Landquart and Lostallo.
lines 99 - 101: after centrifugation, was the supernatant transferred to a clean tube before mixing with binding buffer.
line 111 - 118: briefly described artic v3 vs artic v4.1 protocols. It is not clear if the description provided is only for v4 protocol and how do the two protocols differ. Correct the font for "v3" it should be capital "V3"
line 125: correct 'voc' to 'VOC'
line 129 - 130: using the base count results from V-pipe, how did you identify variant-characteristics mutations in your samples? I understand you have a mutation list. Was the identification done manually or did you use a software?
line 130 - 131: how did you calculate the relative frequency of each signature mutation?
Results. Section 3.1. Variant BA.1 emerged in Landquart on 19th November 2021. A peak can be seen between 3rd and 5th December 2021, then the prevalence decreased by the 23rd December 2021. Those descriptions are missing from section 3.1.
Section 3.3. It would be nice to have a table that records the prevalence of the detected variants (in numbers) as mentioned in the text.
Section 3.2. It would be nice to superimpose the graph of the number of active cases to the regression curve of the SARS-CoV-2 variants for each location. This would help with better visual comparison.
section 3.2. The fluctuation in the number of active cases in Davos and S-chanf, mirrors that of prevalence of BA.1 in wastewater. What about Landquart? The number of active cases recorded for Landquart was higher than Davos and S-chanf. Is there any sequencing data on the identity of the variants that were responsible for the recorded active cases in all 4 catchment areas. That would strengthen the results observed in wastewater samples. In addition that would explain what look like a BA.1 outbreak in Landquart.
Section 3.3. prevalence of BA.5 started increasing a week before WEF began. Has the region witnessed influx of people prior to the meeting?
Section 3.4. line 243: XBB1.5should be corrected to XBB.1.5
line 255 - 256: list SARS-CoV-2 variants that were circulating worldwide 2 weeks before WEF 2023 to 2 weeks after the meeting and provide references; particularly in the regions from where the participants are coming from. That data should be added to Table 1 and S1.
Line 258. edit title of table to let readers know that these are participants to the WEF 2023. Without identity of SARS-CoV-2 variants from the regions, table 1 add little value to study.
Discussion. The evidences presented are weak to conclude regarding the validity of wastewater-based sequencing data. to strengthen your discussion, you need to compare the identity of SARS-CoV-2 variants (as obtained from clinical sequencing data) to the wastewater sequencing data obtained from the catchment areas. With regard to WEF 2022 and 2023 identity of circulating variants in regions from where participants are coming from should be compared to the wastewater-based sequencing results from Davos and S-chanf. Last but not least, in Landquart, the observed high number of active cases, &, rise and fall in the prevalence of BA.1 should be briefly discussed
Comments on the Quality of English LanguageThe manuscript is well written. Minor edits and text restructuring were suggested above.
Author Response
Reviewer 1
Comments and Suggestions for Authors
Congratulations on a well-written manuscript. You have a simple and straightforward design for your study. Wastewater-based surveillance is a great method for unbiased community-wide public health surveillance. However, it serves the public better when it complements clinical data. As more data is produced through wastewater-based epidemiology it will become a standalone surveillance system. Your study lacks originality. It is not clear what novel, wet or dry, laboratory methodology was used to produce the results obtained. The manuscript reads like a surveillance report. I have made few comments below to help increase confidence in the results you have presented.
Response: We appreciate the reviewer's thorough evaluation and constructive feedback, which have been instrumental in enhancing the quality of the manuscript.
Introduction. The background given for the VOC and the the description of the sampling area were well written and appropriate to the context of the study. There was not enough background given regarding wastewater-based sequencing in general and of SARS-CoV-2 in particular. The first paragraph of the discussion gives a nice background that helps understand the aim of the study and should be part of the introduction instead.
Response: We have thoroughly reconstructed and rewritten both the introduction and the discussion sections of the manuscript. The paragraph from the discussion section, as mentioned, has been relocated to the introduction section. It now forms an integral part of an extended introduction, providing more background on SARS-CoV-2, viral RNA shedding in feces, wastewater sequencing of viral RNA and variant mapping, all supported by the relevant references (lines 37-62 in the revised manuscript). The first section of the revised discussion paragraph has been expanded to include a discussion regarding the utility of wastewater sequencing in detecting the spread of novel variants, also in comparison to clinical sample sequencing and with the relevant references (lines 316-345 in the revised manuscript).
Please add a reference for the sentence going from line 58 to 60.
Response: We have included the reference to the publication by Jahn et al., (2022) on the COJAC resource to map the amplicon sequencing data to the different variants (in line 89 of the revised manuscript).
Research design. The design of the project is appropriate for the aim of the study. The sampling dates and locations were well chosen. The sampling dates for the 2021 sport events occurred at least 10 days before the event and 10 days after. The fact that you have searched for defining mutations of VOCs instead of consensus sequences increase confidence in your data. You have made used of the ARTIC v3 protocol for the 2021 samples, and, ARTIC v4.1 for 2022 and 2023 samples. Amplicons drop off were observed when using the v3 primer set. Did you compare the two protocols on your wastewater samples before applying it? What novelty does your study bring? With regard to the use of wastewater-based sequencing; sample processing? data analysis?
Response: The objective of the study presented here focused less on method development and more on the early application of established methods for wastewater sequencing. The goal was to monitor the spatial and time-resolved relative prevalence of various SARS-CoV-2 variants, aiming to investigate an epidemiologically important question regarding the role of large events in the spread of novel variants. The sequencing of the wastewater samples was conducted in three different batches corresponding to the respective time intervals. For each batch, the protocol in use at the Functional Genomics Center Zurich (FGCZ) during that period was applied. To emphasize this point, the last section of the discussion now reads as follows (lines 398-405 of the revised manuscript):
[In terms of early implementation of protective measures, the canton of Grisons played an important pioneering role at the beginning of the pandemic. For instance, the cantonal government took proactive measures by canceling the Engadin Ski Marathon and other events on February 27 2020, even before COVID-19 was declared a pandemic on March 11 by the WHO [44]. In retrospect, and taking into account the data presented here, which demonstrate that large events can contribute to the dis-semination of novel virus variants, these early and resolute decisions were well-founded in their aim to contain the outbreak of the pandemic within a population lacking immunity to SARS-CoV-2.]
Methodology. Well described. Please revise the following:
The back and forth use of S-chanf and St-Moritz between the text and the graphs is confusing. I understand that St-Moritz is located in the catchment area of S-chanf. Choose one and keep it uniform throughout the text and figures to facilitate comparison. Given that sampling is done at S-chanf, but the events happened at St-Moritz description can be given in the introduction or sample collection then S-chanf kept throughout the text and on figures.
Response: We agree with the reviewer that the back and forth between S-chanf and St. Moritz is confusing. We therefore explain the distinction between the place of the event and of the wastewater treatment plant in the introduction (line 113) and in the results section (line 241), but kept S-chanf throughout the remainder of the text and in the Figures. In the Abstract, however, we decided to not make this distinction and to use St. Moritz consistently.
line 91: you mentioned collection from three wastewater treatment plants, but cited 4, i.e. S-chanf, Davos, Landquart and Lostallo.
Response: We thank the reviewer for spotting this mistake, which was corrected in the revised version of the manuscript (line 119).
lines 99 - 101: after centrifugation, was the supernatant transferred to a clean tube before mixing with binding buffer.
Response: The supernatant was transferred to a clean 100 ml glass bottle before mixing with binding buffer. To describe this more clearly, the text now reads as follows (lines 127-129 of the revised manuscript):
After centrifugation at 3000 x g for 10 min at RT, the supernatant was transferred to a glass bottle and gently mixed with 12 ml of binding buffer 1 and 1 ml of binding buffer 2.
line 111 - 118: briefly described artic v3 vs artic v4.1 protocols. It is not clear if the description provided is only for v4 protocol and how do the two protocols differ. Correct the font for "v3" it should be capital "V3"
Response: As previously mentioned, our study did not focus on establishing or comparing sequencing protocols. Instead, we utilized the sequencing protocols in use at the Functional Genomics Center Zurich (FGCZ) during the respective testing periods. Therefore, we prefer to direct the reader to the provided references for the ARTIC V3 and V4.1 protocols. The capital V3 was corrected.
line 125: correct 'voc' to 'VOC'
Response: As the folder name is in lowercase, this was kept to avoid confusion
line 129 - 130: using the base count results from V-pipe, how did you identify variant-characteristics mutations in your samples? I understand you have a mutation list. Was the identification done manually or did you use a software?
line 130 - 131: how did you calculate the relative frequency of each signature mutation?
Response: We apologize for the confusing description. The calculation of the relative frequencies was done using COJAC, which is part of the V-pipe pipeline. The revised description in the methods section makes this clearer and includes the reference to COJAC in which the procedure is explained in more detail. This part of the methods now reads as follows (lines 159 and 160 of the revised manuscript):
[In the COJAC section of V-pipe, a tally of mutation occurrences is done based on the signatures provided [15]. In brief, … ]
Results. Section 3.1. Variant BA.1 emerged in Landquart on 19th November 2021. A peak can be seen between 3rd and 5th December 2021, then the prevalence decreased by the 23rd December 2021. Those descriptions are missing from section 3.1.
Response: This early peak of BA.1 in Landquart is now mentioned and put into context in the revised manuscript. Considering the clinical sequencing data this early peak most probably represents an artifact of the mapping procedure. We therefore choose not to explicitly categorize this as a result but rather to illustrate it in Figure 1 and in the new Supplemental Table S3, and then make a reference to it in the discussion. The corresponding part in the discussion reads as follows (lines 358-364 of the revised manuscript):
[In the clinical sample sequencing data according to CoVariants [33], BA.1 was first detected in the canton of Grisons and Ticino in the time period between 22 November and 6 December 2021 with a frequency of 1.2%. In view of these data, the early peak of BA.1, which reached a relative prevalence of 10.4% on 26 November 2021 in Landquart in the data presented here (Figure 1, Table S3), is most probably an artifact of the mapping procedure rather than a genuine spread of BA.1.]
Section 3.3. It would be nice to have a table that records the prevalence of the detected variants (in numbers) as mentioned in the text.
Response: The result tables with the point estimates of the relative frequencies for each variant, as well as their upper and lower confidence limits are now provided for each test period and location in Supplementary Tables S2-S5 and S7-S10.
Section 3.2. It would be nice to superimpose the graph of the number of active cases to the regression curve of the SARS-CoV-2 variants for each location. This would help with better visual comparison.
Response: The attempts to superimpose the graphs resulted in highly cluttered figures. Therefore, we have chosen to retain the original figure designs.
Section 3.2. The fluctuation in the number of active cases in Davos and S-chanf, mirrors that of prevalence of BA.1 in wastewater. What about Landquart? The number of active cases recorded for Landquart was higher than Davos and S-chanf. Is there any sequencing data on the identity of the variants that were responsible for the recorded active cases in all 4 catchment areas. That would strengthen the results observed in wastewater samples. In addition that would explain what look like a BA.1 outbreak in Landquart.
Response: There are unfortunately no sequencing data of clinical samples specifically for the four catchment areas available. However, in the CoVariants resource the identity and frequency of the strains identified in clinical samples are provided for different regions in Switzerland, including Region 6, which encompasses the four testing sites included here. This information has now been included in the manuscript (Supplementary Table S6 and text in lines 212-219 in the revised manuscript):
[The arrival of Omicron BA.1 was also monitored in the clinical sample sequencing data according to CoVariants [33]. The strain identified in all 322 clinical samples sequenced during the period of 8 and 22 November 2021 in Region 6 of Switzerland, encompassing the four testing sites included here, remained the Delta variant. In the period between 22 November and 6 December 2021, 3 out of 252 sequenced strains were Omicron BA.1 (21K), which increased to 76 out of 242 sequenced samples between 6 and 20 December 2021. In the next period between 20 December 2021 and 3 January 2022, the frequency of Omicron BA.1 was already increased to 86% (Table S6).]
As mentioned above, these data additionally put the early peak of BA.1 in November 2021 in question. Therefore, we suggest that the high number of active cases recorded in Landquart towards the end of November was caused by the Delta variant. This high number of cases was subsequently contained through the successful testing and isolation strategy that was still successfully applied at that time.
Section 3.3. prevalence of BA.5 started increasing a week before WEF began. Has the region witnessed influx of people prior to the meeting?
Response: Already before the actual start of the WEF, there is usually a large influx of people, which can explain the increasing prevalence of BA.5.
Section 3.4. line 243: XBB1.5should be corrected to XBB.1.5
Response: This was corrected.
line 255 - 256: list SARS-CoV-2 variants that were circulating worldwide 2 weeks before WEF 2023 to 2 weeks after the meeting and provide references; particularly in the regions from where the participants are coming from. That data should be added to Table 1 and S1.
Response: As shown in the map below from CoVariants for January 2023, the information on SARS-CoV-2 variants circulating worldwide is very complex. We therefore chose to focus on the worldwide distribution solely of BA.2.75 between 2 and 9 January 2023. For this time period, the frequencies of BA.2.75 in the 20 countries with the highest frequencies are given now in new Table 1 (the previous Table 1 now is Table 2). These data contrast with the countries of origin from which the official participants of the WEF came from. This strengthens our point that large international events can promote the spread of new variants in the respective host region.
Line 258. edit title of table to let readers know that these are participants to the WEF 2023. Without identity of SARS-CoV-2 variants from the regions, table 1 add little value to study.
Response: The title of Table 2 of the revised manuscript (was Table 1 in the original version) now reads as follows to make clear that these are participants to the WEF 2023 (line 208 of the revised manuscript):
[The official number of WEF 2023 participants per region of origin.]
The new Table 1 presents the frequencies of BA.2.75 in countries with the highest frequencies of this variant. This emphasizes that the major international event contributed to the spread of the variant in the host region of WEF 2023, where it had been virtually absent previously.
Discussion. The evidences presented are weak to conclude regarding the validity of wastewater-based sequencing data. to strengthen your discussion, you need to compare the identity of SARS-CoV-2 variants (as obtained from clinical sequencing data) to the wastewater sequencing data obtained from the catchment areas.
Response: The clinical sequencing data for Region 6 of Switzerland have been included for November and December 2021 (already mentioned above) and for December 2022 and January 2023 (Supplemental Table S11 and text in lines 293-297 of the revised manuscript). It is worth to mention that in the comparable time intervals, only about 1/10 of the samples were sequenced in 2022/2023 compared to 2020/2021, which further emphasizes the value of the wastewater sequencing data. A thorough discussion on various wastewater sequencing efforts in comparison to clinical genome surveillance was included at the first part of the introduction concluding with the following statement (lines 341 - 345 of the revised manuscript):
[These studies have highlighted the ability of wastewater sequencing to identify circulating SARS-CoV-2 variants within defined geographical regions delineated by the catchment area of the respective wastewater treatment plants, either independently or in addition to clinical genomic surveillance data.]
With regard to WEF 2022 and 2023 identity of circulating variants in regions from where participants are coming from should be compared to the wastewater-based sequencing results from Davos and S-chanf.
Response: As mentioned above, a new table has ben included with the frequencies of BA.2.75 for the countries with the highest frequencies at the beginning of January 2023. .
Last but not least, in Landquart, the observed high number of active cases, &, rise and fall in the prevalence of BA.1 should be briefly discussed
Response: As mentioned above, a discussion on the rise and fall in the prevalence of BA.1 in November 2021 has been included.

Reviewer 2 Report
Comments and Suggestions for Authors
Zhakparov et al. presents a very timely report of the application of wastewater surveillance to analyze and characterize the prevalence of SARS-CoV-2 variants in different regions of Switzerland and compare them in the context of their exposure to international events. The methods are well described and the conclusions are well supported. Below a couple of personal suggestions, but other than that the report is very straightforward.
1. To justify the conclusion in the abstract: "We can therefore conclude that major international events promote the spread of new variants in the respective host region", I suggest making clearer that the increase on cases and higher prevalence of BA.1 are compared against the Landquart and Lostallo regions, that I am assuming work as controls not exposed to the international events. Otherwise the value of the contrast could be missed to the readership.
2. Introduction, line 42 reads: "The following period was characterized by a high rate of evolution due to an increase in divergence", what does 'high rate of evolution' mean in this context? I suggest to change it for 'adaptations to human host' or 'transmissibility'; but evolution (it is a continuous process) and a 'high rate' indicates something that can be measured, in which case 'mutation rate' could be more appropriate.
Author Response
Reviewer 2
Comments and Suggestions for Authors
Zhakparov et al. presents a very timely report of the application of wastewater surveillance to analyze and characterize the prevalence of SARS-CoV-2 variants in different regions of Switzerland and compare them in the context of their exposure to international events. The methods are well described and the conclusions are well supported. Below a couple of personal suggestions, but other than that the report is very straightforward.
Response: We thank the reviewer for the positive assessment and constructive feedback.
- To justify the conclusion in the abstract: "We can therefore conclude that major international events promote the spread of new variants in the respective host region", I suggest making clearer that the increase on cases and higher prevalence of BA.1 are compared against the Landquart and Lostallo regions, that I am assuming work as controls not exposed to the international events. Otherwise the value of the contrast could be missed to the readership.
Response: To emphasize our point that international events contribute to the spread of new variants only within their host regions, we have now included comparisons with the respective control regions. This section of the abstract now reads as follows:
[The prevalence of the variants identified from the wastewater sequencing data showed that the Omicron variant BA.1 had spread in Davos and St. Moritz during the international sporting events hosted there. This spread was associated with an increase in case numbers, while it was not ob-served in Landquart and Lostallo. Another instance of new variant spread occurred during the WEF in January 2023, when the Omicron variant BA.2.75 arrived in Davos but not in St. Moritz.]
- Introduction, line 42 reads: “The following period was characterized by a high rate of evolution due to an increase in divergence”, what does ‘high rate of evolution’ mean in this context? I suggest to change it for ‘adaptations to human host’ or ‘transmissibility’; but evolution (it is a continuous process) and a ‘high rate’ indicates something that can be measured, in which case ‘mutation rate’ could be more appropriate.
Response: We thank the reviewer for pointing this out. This paragraph has been slightly expanded and reformulated to make this point clearer. The text now reads as follows (introduction, lines 67-73):
[The first three divergent lineages with a high number of mutations emerging in different regions of the world were Alpha, Beta and Gamma. These distinct lineages exhibit some convergent mutations, implying that they confer a fitness advantage in evading population immunity, which subjects the virus to selective pressure. The following period was characterized by a pronounced evolutionary diversification marked by a gradual increase in divergence within the major lineages combined with a stepwise increase as new major lineages emerged [5–7].]

Reviewer 3 Report
Comments and Suggestions for Authors
Presented manuscript describes sequencing of SARS-CoV-2 RNA fragments isolated form wastewater in order to study the spread of new variants during various events with the participation of a large number of people.
In general, these data are of some interest. Nevertheless, the manuscript is hard to read. Despite the large number of drawings, they poorly structure information about genotypes, sampling sites and ongoing events. I would like to see a more statistically sound analysis of correlations of an increase or decrease in specific varaints, including population sizes and frequencies, and the specifics of testing. Has the DNA of human bacterioids been determined, for example, as a criterion for assessing the load of human faeces? These data would make the analysis more objective. I would like to see a more clearly formulated conclusion and limitations of the study. I suppose the idea concerning the higher infectivity of Omicron BA.1 compared to the wildtype strain is not so new.
The authors do not describe or discuss similar studies conducted in other countries.
Author Response
Reviewer 3
Comments and Suggestions for Authors
Presented manuscript describes sequencing of SARS-CoV-2 RNA fragments isolated form wastewater in order to study the spread of new variants during various events with the participation of a large number of people.
In general, these data are of some interest. Nevertheless, the manuscript is hard to read. Despite the large number of drawings, they poorly structure information about genotypes, sampling sites and ongoing events.
Response: We thank the reviewer for the valuable feedback. In response to the reviewer's comments, we have undertaken a thorough revision of the manuscript, which includes restructuring the sections of the text and incorporating additional information. We believe these changes have resulted in an improved manuscript with enhanced clarity and readability.
I would like to see a more statistically sound analysis of correlations of an increase or decrease in specific varaints, including population sizes and frequencies, and the specifics of testing. Has the DNA of human bacterioids been determined, for example, as a criterion for assessing the load of human faeces? These data would make the analysis more objective.
Response: In the thoroughly revised introduction of this manuscript, we introduce the different methods for analyzing wastewater samples. These methods include quantitative PCR for assessing viral gene counts and amplicon sequencing of the viral RNA to determine the relative prevalence of different virus strains. The first method has previously been successfully applied in the canton of Grisons, except for the time when Omicron BA.1 first arrived. In that project, we developed a normalization strategy to account for fluctuating population levels and varying wastewater loads, and we attempted to identify a factor that would enable us to make the results from the initial PCRs comparable to those that were adjusted to incorporate Omicron as well. However, this is not part of the data presented here. The second method, amplicon sequencing to determine the relative prevalence of different viral strains, remains susceptible to variations in sample quality. Nevertheless, it is not dependent on wastewater load as long as sufficient viral RNA can be extracted. We hope that the revised text clarifies this point.
The method that was applied to obtain the relative prevalence data presented in this manuscript was V-pipe, which is based on statistical modeling to detect and quantify the presence of SARS-CoV-2 variants. To assess the prevalence of every variant we utilized smoothing regression in the earlier version and kernel-based deconvolution in the current version, and have added the corresponding literature references. The determination of the relative frequencies therefore heavily relies on statistics. In the revised manuscript we have included the results tables of V-pipe in Tables S2-S5 and S7-S10, which include the point estimates, as well as the upper and lower confidence limits. The incorporation of these tables and the new section in the discussion that point out how this method can be misled should underscore that these findings are a product of statistical analysis.
I would like to see a more clearly formulated conclusion and limitations of the study.
Response: The conclusion paragraph has been thoroughly revised. The current version presents clearer conclusions, for instance regarding the retrospective validation of the early measures taken by the cantonal government to contain the pandemic and prevent the strain on the medical system, as observed at that time in neighboring Northern Italy. The revised version also underscores the value of the wastewater sequencing data in periods when clinical genomic surveillance is reduced. To address the limitations, we have included a section explaining the principle of how V-Pipe works and highlighted that it can give misleading results when dealing with lower-quality samples or when the variant mapping file does not accurately represent the strains circulating at the time and in the region when the wastewater samples were collected.
I suppose the idea concerning the higher infectivity of Omicron BA.1 compared to the wildtype strain is not so new.
Response: We of course agree with this view. In addition to referencing the higher infectivity of Omicron BA.1 in the introduction and in section 3.2 of the results (line 246), we have included the references now also in the discussion (line 377 of the revised manuscript).
The authors do not describe or discuss similar studies conducted in other countries.
Response: In the thoroughly revised conclusions, we have incorporated a paragraph discussing other studies on wastewater sequencing of SARS-CoV-2 RNA conducted in various countries. This paragraph now reads as follows (lines 316 - 345 of the revised manuscript):
[The method of wastewater sequencing of SARS-CoV-2 RNA and determination of variant prevalence has been widely adopted worldwide. Even at the beginning of the pandemic, when wild-type SARS-CoV-2 was spreading, wastewater sequencing was used to assess viral diversity in samples from 11 states in the USA and from 4 different wastewater treatment plants in the San Francisco Bay Area, and it was found that the wastewater data identified more circulating lineages than represented in the clinical data [17,39]. Comparing the results acquired using RT-PCR screening and genome sequencing in wastewater and using patient sample sequencing in Marseille, France, has revealed that sequencing allows for detecting a distribution of variants in wastewater that is similar to patient sample sequencing, while some variants are missed by RT-PCR [40]. In Spain, sequencing of SARS-CoV-2 RNA in wastewater from 14 wastewater treatment plants during three epidemiological waves allowed for the detection of the Alpha variant and showed the capacity of the method to detect mutations before they are detected in clinical samples [41]. Also in Switzerland the Alpha variant was detected in wastewater up to 13 days before it was first reported in clinical samples, and wastewater sequencing was found to provide population-level estimates for the prevalence of emerging variants earlier and based on fewer samples than based on clinical samples [15]. The wastewater monitoring initiative at the University of California San Diego (UCSD) campus in the USA also revealed that emerging VOCs could be identified up to 14 days earlier in wastewater samples in comparison to clinical genomic surveillance. Furthermore, it was observed that instances of virus trans-mission could be detected in wastewater samples that had not been identified through clinical testing [21]. Furthermore, a pan-European wastewater-based SARS-CoV-2 surveillance comparing the mutation profiles of Alpha, Beta, Gamma and Delta in 54 European municipalities has revealed that data on SARS-CoV-2 VOCs in wastewater samples mirror variant profiles obtained from clinical data [42]. These studies have highlighted the ability of wastewater sequencing to identify circulating SARS-CoV-2 variants within defined geographical regions delineated by the catchment area of the respective wastewater treatment plants, either independently or in addition to clinical genomic surveillance data.]

Round 2
Reviewer 1 Report
Comments and Suggestions for Authors
Thank you for revising the manuscript. It reads much better. Please revise the following:
Results.
Figures 1, 3 and 4 have extra border lines that were not in the previous version. Please remove them.
section 3.1.lines 215 - 219 of revised manuscript: to describe clinical cases due to BA.1 choose either number of sequences (i.e. from 3 out of 252 sequences, ... 76 out of 242 sequences and 66 out of 477 sequences) or frequency (from <1%, ...to 31%, ...to 86%). By changing from number of sequences to frequency in line 218 and 219 it can get confusing.
Table S6. Thank you for adding it, it illustrates well the point you are making. can you please add it to text instead of supplementary material.
Section 3.3. line 267. revise "... monitored the arrival of BA.5 in S-chanf" to "...monitored the potential arrival of BA.5 in S-chanf". with wastewater data you need at least 2 consecutive weeks of data to make sure the result you have is not an artefact. Given that the relative frequency of BA.5 in S-chanf increased from <1% on the 15 May 2022 to ~35% on the 19 May 2022 and you don't have data prior to 15 May 2022 it is safer to use 'potential' to describe the arrival of BA.5 in S-chanf.
Section 3.4. include Table S11 in the manuscript and not supplemental material
Discussion. Lines 362 - 363. I don't agree that the relative frequency recorded for Landquart could be an artefact of mapping. Table S3 clearly records 3 point estimates (i.e. 7.2% - 10.4% - 6.9%) (the first two represent two consecutive weeks) that suggest increase and decrease in the prevalence of BA.1 in Landquart. asymptomatic cases could have accounted for the results. consider revising those lines.
is there a next step after your study? if so add details in the last paragraph of the discussion.
Author Response
Reviewer 1 – Round 2
Thank you for revising the manuscript. It reads much better.
Response: We thank the reviewer again for the constructive feed-back and for agreeing that the revisions have enhanced the quality of the manuscript.
Please revise the following:
Results.
Figures 1, 3 and 4 have extra border lines that were not in the previous version. Please remove them.
Response: These lines are no longer visible in the final versions of the Figures.
section 3.1.lines 215 - 219 of revised manuscript: to describe clinical cases due to BA.1 choose either number of sequences (i.e. from 3 out of 252 sequences, ... 76 out of 242 sequences and 66 out of 477 sequences) or frequency (from <1%, ...to 31%, ...to 86%). By changing from number of sequences to frequency in line 218 and 219 it can get confusing.
Response: We agree with the reviewer that this can get confusing. The text has therefore been changed and reads as follows (lines 218 – 219 of revision 2 of the manuscript):
[In the next period between 20 December 2021 and 3 January 2022, already 410 of 477 sequenced strains were BA.1, reaching a frequency of 86% (Table 1).]
Since the former Table S6 is now Table 1 and is part of the main text, this information is now also directly accessible.
Table S6. Thank you for adding it, it illustrates well the point you are making. can you please add it to text instead of supplementary material.
Section 3.4. include Table S11 in the manuscript and not supplemental material
Response: As mentioned above, the former Table S6 is now Table 1 of the main text, and the former Table S11 is now Table 3. For better comparison, the other names of the variants mainly used in the text have been included in the new Tables 1 and 3.
Section 3.3. line 267. revise "... monitored the arrival of BA.5 in S-chanf" to "...monitored the potential arrival of BA.5 in S-chanf". with wastewater data you need at least 2 consecutive weeks of data to make sure the result you have is not an artefact. Given that the relative frequency of BA.5 in S-chanf increased from <1% on the 15 May 2022 to ~35% on the 19 May 2022 and you don't have data prior to 15 May 2022 it is safer to use 'potential' to describe the arrival of BA.5 in S-chanf.
Response: We thank the reviewer for the careful consideration and have added ‘potential’ to line 270 of revision 2 of the manuscript.
Discussion. Lines 362 - 363. I don't agree that the relative frequency recorded for Landquart could be an artefact of mapping. Table S3 clearly records 3 point estimates (i.e. 7.2% - 10.4% - 6.9%) (the first two represent two consecutive weeks) that suggest increase and decrease in the prevalence of BA.1 in Landquart. asymptomatic cases could have accounted for the results. consider revising those lines.
Response: We have re-evaluated and discussed the data in detail, taking into account the relative prevalence data of the other variants in Landquart in late November and early December 2021, and re-assessing the qPCR wastewater data (now provided as Supplementary Figure S1). We arrived at the conclusion that both options are possible, although we lean towards it being an artifact of the quantification procedure. This part of the discussion has been reformulated to account for this and now reads as follows (lines 365 and 372 of revision 2 of the manuscript):
[In view of these data, the early peak of BA.1, which reached a relative prevalence of 10.4% on 26 November 2021 in Landquart in the data presented here (Figure 1, Table S3), might be an artifact of the mapping procedure. Further supporting this view are the additional peaks in the relative prevalences of the Kappa and Alpha variants, as well as in the undetermined fraction, which was only observed in the Landquart samples during those dates. These findings suggest a potential lower data quality of these samples, which could contribute to the overdispersion effect. The quantification section of the V-Pipe pipeline… ]
is there a next step after your study? if so add details in the last paragraph of the discussion.
Response: Wastewater monitoring as a routine practice has unfortunately been discontinued in the canton by the end of June 2023. Still, in view of the data presented here, we are currently planning to resume it during future large international events. The following sentence was therefore added to the discussion (lines 405-406 of revision 2 of the manuscript):
[We therefore intend to maintain wastewater monitoring during future large international events, despite its discontinuation here as a routine practice.]
